# Cognitive Function Decline in the Third Trimester of Pregnancy Is Associated with Sleep Fragmentation

**DOI:** 10.3390/jcm11195607

**Published:** 2022-09-23

**Authors:** Dorota Wołyńczyk-Gmaj, Aleksandra Majewska, Aleksandra Bramorska, Anna Różańska-Walędziak, Simon Ziemka, Aneta Brzezicka, Bartłomiej Gmaj, Krzysztof Czajkowski, Marcin Wojnar

**Affiliations:** 1Department of Psychiatry, Medical University of Warsaw, Nowowiejska 27, 00-665 Warsaw, Poland; 2Department of Psychology, University of Social Sciences and Humanities, Chodakowska 19/31, 03-815 Warsaw, Poland; 3Department of Human Physiology and Patophysiology, Faculty of Medicine, Collegium Medicum, Cardinal Stefan Wyszynski, 01-938 Warsaw, Poland; 4II Department of Obstetrics and Gynecology, Medical University of Warsaw, Karowa 2, 00-315 Warsaw, Poland

**Keywords:** working memory, awakenings, pregnancy, attention, actigraphy

## Abstract

During late pregnancy, sleep deterioration is regularly observed. In concert with these observations, in previous studies by other researchers, a slight objective cognitive decline in pregnant women has been found. Sleep is essential for memory consolidation. The hypothesis of the study was that cognitive impairment could be related to sleep deterioration during pregnancy. The study included 19 pregnant women in their third trimester of pregnancy (28–40 weeks, median 33 weeks (IQR 32–37)) recruited at the Department of Gynecology and Obstetrics, Medical University of Warsaw, and 20 non-pregnant women as controls. The assessment was performed using the vocabulary subtest from the Wechsler Adult Intelligence Scale (WAIS), D2 Test of Attention, OSPAN task (Operational Span Task) to assess cognitive performance, actigraphy to examine sleep parameters, and a set of self-report instruments: Athens Insomnia Scale (AIS), Beck Depression Inventory (BDI), Ford Insomnia Response to Stress (FIRST), Regenstein Hyperarousal Scale (HS), and Epworth Sleepiness Scale (ESS). Although there were no differences between the groups in WAIS (*p* = 0.18), pregnant women had worse scores in working memory capacity (overall number of remembered letters: *p* = 0.012, WM span index: *p* = 0.004) and a significantly lower score in attention (*p* = 0.03). Pregnant women also had lower sleep efficiency (*p* = 0.001), more awakenings from sleep (*p* = 0.001), longer average awakenings (*p* < 0.0001), longer wake after sleep onset (WASO, *p* < 0.0001), and longer total time in bed (*p* < 0.0001). In psychological assessment, pregnant women had only a higher FIRST score (*p* = 0.02). Using mediation analysis, we found that frequent awakening might be the major factor contributing to deterioration in working memory performance, explaining almost 40% of the total effect. In conclusion, sleep fragmentation in the third trimester of pregnancy may impair working memory consolidation. Pregnant women often complain about poor daily performance as well as non-restorative sleep. In this study, we showed that there is a relationship between lower sleep quality in pregnancy and worse cognitive functioning. We can expect a cognitive decline in women with sleep disturbances in pregnancy. Therefore, we should pay more attention to the treatment of sleep disorders in pregnancy.

## 1. Introduction

### 1.1. Sleep and Cognition in the Third Trimester of Pregnancy

During pregnancy, the majority of pregnant women claim of cognitive deterioration [1,2]. Thus, there are contradictory research results about objective cognitive impairment in pregnancy. Sleep disorders are also frequent complaints in pregnant women. Insomnia symptom prevalence growing in the subsequent months of pregnancy, together with typical pregnancy, affects 52.5–63% of pregnant women depending on the adopted criteria for insomnia in the third trimester, a figure much higher than in the general population [3,4,5]. In addition, pregnant women are more likely to suffer from restless leg syndrome (RLS) and obstructive sleep apnea (OSA), with the prevalence of both increasing to 23% and 27% respectively at the end of pregnancy [6,7]. Thus, both cognition and sleep are reported to be impaired in pregnancy.

### 1.2. Relationship between Pregnancy and Cognitive Function—Review of the Literature

De Groot et al., (2006) have found an adverse effect of late pregnancy on verbal memory [8]. Buckwalter et al., (1999) showed that the memory impairment during pregnancy identified in the research is not related to mood impairment or hormonal levels [9]. Another study conducted on naturalistic tasks, closely representing those occurring in daily life, revealed more errors in prospective memory tests in pregnancy [10]. Farrar et al. [11] found worse Cambridge Neuropsychological Automated Test Battery (CANTAB) results of spatial recognition memory in women in the second trimester of pregnancy and postpartum as compared to the same group in the first trimester of pregnancy and a non-pregnant control group.

Results of meta-analysis of 14 studies published between 1991 and 2007 comparing pregnant and/or postpartum women with healthy matched controls indicate impairment of high demand executive cognitive functions in pregnancy [12]. The primiparous women (those pregnant for the first time) could be more prone to implicit memory deterioration during pregnancy than multiparous women [13].

Thus, some research found no significant differences in cognitive tests between pregnant and non-pregnant women, even when participants reported cognitive problems [14,15]. A study including 21 women in the second and third trimesters of pregnancy revealed only worse verbal fluency, which was clinically insignificant [16].

The results of the studies of cognitive functioning during pregnancy could vary depending on the length of the tasks. In the short tasks, pregnant women could more easily overcome cognitive difficulties [17]. Other factors that affect cognition and may cause inconsistent results in pregnant women are depression and anxiety [18].

Cognitive decline in pregnancy may be related to brain changes in pregnant first-time mothers compared to control non-pregnant nulliparous women in such regions as: frontal cortex, temporal cortex, insula, orbitofrontal cingulate cortex, left hippocampus, and parahippocampal gyrus cortex [19]. The pregnancy-dependent gray matter reductions—apart from partial hippocampal volume—endured for at least 2 years after giving birth [19].

### 1.3. Sleep and Pregnancy—Review of the Literature

Apart from subjective complaints of pregnant women, objective studies also show a deterioration in the quality of sleep during pregnancy. It has been established in many studies that the quality of sleep is worse in pregnancy, especially in the third trimester [20,21]. According to a study by Wilson et al. [22], there is more WASO (wake after sleep onset), time of slow-wave sleep (SWS) is shorter, and time of a rapid eye movement (REM) sleep, and non-rapid eye movement (NREM) sleep is longer in the third trimester of pregnancy as compared to non-pregnant women.

A Neau et al., study conducted on 871 women found that subjective vigilance problems could be associated with sleep deterioration during pregnancy [3]. Both deterioration of cognitive function (subjective and objective) and deterioration of sleep quality seem to be more pronounced in women in their late pregnancy.

### 1.4. Memory and Sleep Quality

Memory function is related to the quality of sleep [23]. During neuronal activity of SWS hippocampus-dependent memories are transmitted to the neocortex. During REM sleep, the synaptic consolidation of memories in the cortex occurs [24]. Moreover, acute sleep loss, sleep restriction, and sleep fragmentation all have a negative impact on the consolidation of newly learned information [25]. Furthermore, patients with sleep disorders such as insomnia and sleep apnea suffer from impairments of sleep-dependent consolidation for verbal declarative information [26]. An insomnia meta-analytic review reported mild deficits of working memory in insomnia subjects compared to controls [27]. Sleep disorders may also exacerbate cognitive problems in the elderly through impairment of sleep-dependent memory consolidation processes [28]. Individuals with sleep problems—including short and long sleep duration, poor sleep quality, circadian rhythm abnormality, insomnia, and obstructive sleep apnea (OSA)—had 1.68-times higher risk for the combined outcome of cognitive decline and Alzheimer’s disease compared to individuals without sleep problems [29].

The effect of sleep fragmentation on cognition seems to be similar to the effect of sleep deprivation [30]. In everyday functioning, chronically sleep-deprived subjects have a higher number of driving accidents compared to normal sleepers [31].

Since sleep is essential for proper memory function, and pregnant women reported both sleep and memory disorders, we decided to investigate the effect of objectively measured sleep quality on cognitive task scores in the group of women in the third trimester of pregnancy.

That is why we created the mediation model, where we treat sleep quality as a potential mediator in the relationship between pregnancy and cognitive functioning.

The aim of the current study was to test a hypothesis that sleep deterioration during pregnancy is associated with cognitive function decline. We examined sleep quality and cognitive functions (working memory and attention) in pregnant women as compared to a control group of non-pregnant women.

## 2. Materials and Methods

A total of 41 women of childbearing age (21–45 years) were enrolled in the study. Among them, there was a study group of 21 women in the third trimester of a normal course pregnancy who subsequently came for a routine control visit to the II Department of Gynecology and Obstetrics, Medical University of Warsaw between September 2014 and April 2015 and 20 non-pregnant women as a control group. All subjects gave their informed written consent before completing the questionnaires. The study protocol was approved by the Bioethics Committee of the Medical University of Warsaw (KB/254/2012).

Exclusion criteria included chronic medical and psychiatric disorders or pregnancy-related complications such as preeclampsia and cervical insufficiency.

The final dataset was completed for groups of 19 pregnant and 20 non-pregnant women, respectively. The study and control group matched in terms of age (mean: 31.1 ± 5.4 vs. 30.8 ± 5.34, *p* = 0.84) and education (higher ed: 17 (81%) vs. 15 (75%), *p* = 0.64).

In the study, women’s pregnancy duration was between 28 and 40 weeks (median: 33 (IQR 32–37)). Pregnant women had higher weight (mean 75.7 ± 11.1 kg vs. 61.5 ± 10. 10.64 kg, *p* = 0.0002, t = 4.1) and body mass index (BMI, 27.49 ± 4.2 vs. 22.08 ± 3.8, *p* = 0.0002, t = 4.1). Pregnant women were also more often married (16 vs. 6, *p* = 0.03, chi-square = 11.8). The majority of women in both groups declared being professionally active (17 pregnant and 17 non-pregnant women, *p* = 0.34).

Among all participants, three women worked on night shifts (one pregnant and two non-pregnant women). In terms of medical problems, three women were treated for hypothyroidism (two non-pregnant, one pregnant woman), one pregnant woman for hypertension, and one pregnant—for asthma. Among pregnant women, nine avoided coffee consumption and seven in the control group (*p* = 0.43). Six pregnant women complained about legs tingling.

## 3. Instruments and Procedures

### 3.1. Cognitive Performance

To measure cognitive ability, we used the vocabulary subtest of the Wechsler Adult Intelligence Test (The Verbal Comprehension Index; WAIS-III), the d2 Test of Attention, and the OSPAN (Operational Span Task) task. The participants performed tests at the clinic after the control visit or at home. Each session lasted approximately one hour.

Wechsler Adult Intelligence Test (The Verbal Comprehension Index; WAIS-III)—The vocabulary subtest from WAIS-III measures expressive vocabulary and verbal knowledge that is purported to be a good estimate of crystallized intelligence and general intelligence. Subjects were expected to verbally define 33 words (nouns, verbs, adjectives, adverbs) from the list. The test was interrupted if the participant gave five consecutive incorrect answers in describing words [32].The D2 Test of Attention—a cancellation test of attention and concentration, is a neuropsychological test designed to measure processing speed, rule compliance, and quality of performance, allowing for a neuropsychological estimation of individual attention and concentration performance. Its performance depends on a combination of visual, motor, and attention skills. The test contains 14 lines and participants have 20 s to work on each line in order to cancel out any “d” letter with two marks around the letter. Crucial difficulty in this test is related to distractors, very similar to the target. They are also built with the letters—“d” and “*p*” with marks, but in another arrangement than the target. Three indices of task performance were computed: concentration performance (CP), defined as the number of ‘hits’ (marked targets) minus the number of distractors, which were marked (errors of commission). The CP score is a measure of processing speed adjusted for errors made. Processed targets (PT), defined as the number of target symbols in the ‘processed’ portion of the test up to and including the last response marked on each screen. It equals the number of ‘hits’ (targets found) plus the number of overlooked targets. The PT score is a measure of processing speed without consideration of accuracy. Accuracy (%), was defined as the total number of errors (errors of omission and commission) by the number of processed targets (PT) and expressing this fraction as a percentage. This score is then reversed so that a high standard score reflects a highly accurate response [33,34].The OSPAN task (operational span task)—a modified online version (GEX Immergo, Funds Auxilium Sp. z o.o.) of the operation span (OSPAN) task—measures working memory capacity (WMC) and is closely related to other higher-order intellectual functions. Participants were asked to perform simple mathematical verifications while simultaneously trying to remember a series of letters. After a series of practice trials, participants completed 15 trials ranging from three to seven letters in load. Only trials in which all letters were remembered and recalled in correct order were coded as correct, and this absolute OSPAN score was treated as our individual WMC measure [35,36,37].

### 3.2. Sleep Quality Measurement

To measure objectively time and quality of sleep in the natural home environment we used ActiGraph’s *ActiSleep* water-resistant tri-axial accelerometers (ActiGraph; Pensacola, FL, USA) in all participants for one week. The participants were asked to place the ActiGraphs on the non-dominant wrists before going to bed. During the recorded period, subjects were asked to fill the sleep log daily.

These devices are piezoelectric accelerometers with a sensitivity of less than 0.01 g and a sampling rate of 30 Hz. Actigraphic data during 30-s epochs were scored as sleep or wake by Activate-Sleep^®^ v. 2.53 analysis software (Mini Mitter Co., Inc., Bend, OR, USA). Actisleep was found to be a valid and reliable device for sleep measurements when worn on the wrists of healthy children and young adults [38,39]. Study has showed that actigraphy, in comparison to polysomnography, could be a reliable measure in insomnia patients, in terms of the number of awakenings, WASO, total sleep time TST, and SE; however, it failed to estimate SL. We used the Sadeh algorithm, validated in a young population, to score sleep [40]. We set total sleep time (TST; in minutes), sleep efficiency (SE; TST divided by the time the subject was in bed), wake after sleep onset (WASO; time of wake in minutes after sleep onset), number of awakenings and the sleep latency (SL; time to fall asleep in minutes).

Additionally, in the majority of subjects (in 12 pregnant and in 13 non-pregnant women), we checked pulse oximetry parameters for three consecutive nights by means of the wrist oxygen saturation monitor Minolta Pulsox-300i (Konica Minolta). Sleep apnea can negatively affect cognitive function, so we compared oxygen saturation between the groups. The mean oxygen saturation of blood in the pregnant group was 96.86 ± 0.2% and in the control group—95.53 ± 1.9% (t = 0.23, *p* = 0.41). The mean number of dips of oxygen saturations below 4% (5.2 ± 3.9 for pregnant and 8.29 ± 12.9 for healthy control; t = 0.8, *p* = 0.43) and below 3% (5.21 ± 3.9 for pregnant vs. 8.29 ± 12.9 for controls, t = 0.02, *p* = 0.82) did not differ significantly between the samples.

### 3.3. Psychological Measurement

We used instruments to assess sleep problems, depressive symptoms, and hyperarousal:Athens Insomnia Scale (AIS)—An eight-item self-reported questionnaire based on the ICD-10 criteria designed for quantitative measurement of severity of insomnia. Each item is rated from 0 (not a problem) to 3 (a very serious problem) with a total score from 0 to 24. The scale is characterized by a very good consistency (Cronbach’s alpha = 0.90) and reliability (test–retest reliability, r2 = 0.92) [41,42]. AIS is one of the most commonly used scales for diagnostic purposes as well as research on the effectiveness of insomnia treatment. The scale was validated in Poland with 8 points as a cut-off score.Epworth Sleepiness Scale (ESS)—An eight-item self-reported questionnaire with a range from 0 to 24 points, used to determine the level of daytime sleepiness in populations suffering from a variety of sleep disorders [43]. A score ≥10 is usually considered abnormal, indicating excessive daytime sleepiness (EDS). It was assessed in pregnant women and is characterized by a good consistency (Cronbach’s alpha = 0.8) [44].Ford Insomnia Response to Stress Test (FIRST)—A nine-item self-report measure of susceptibility to stress manifested by a deterioration of sleep with four possible answers scored from 1 to 4 points according to the possibility of sleep deterioration after described in items situation. The total score ranges from 9 to 36 [45]. The scale has been tested as reliable for assessing susceptibility to insomnia among women at their early pregnancy [46].Beck Depression Inventory (BDI)—A 21-item self-report scale; responses for each item are scored from 0 to 3 depending on severity of symptoms. The score ranges from 0 to 63 and the cut-off point for the Polish population is 12. BDI is used to assess the severity of depressive symptoms with good internal consistency (Cronbach’s alpha = 0.85) [47,48].Regenstein Hyperarousal Scale (HS)—A 26-item self-reported scale, responses are scored from 0 to 3 with the total score ranges from 0 to 78 points. HS is used to assess hyperarousal with a good consistency (Cronbach’s alpha = 0.84) and is well correlated with objective measures of alertness [49].

In addition, a set of structured questions about common symptoms of sleep disturbances, such as sleep nightmares, snoring, and tingling in the legs (restless leg syndrome—RLS). Moreover, data on social, demographic, and current medical status were collected.

## 4. Statistical Analysis

The data were analyzed using SAS 9.4 software (SAS Institute Inc., Cary, NC, USA) and SPSS Statistics (IBM Corp.). In the first step of exploratory analysis, we tested, with a Student’s *t*-test, for differences between pregnant and non-pregnant women in sleep parameters, psychological scale scores, and cognitive function. In the next step, a series of simple linear regressions (with a single predictor, without covariates) were performed as a preparatory step for mediation analyses. We took the variables with the strongest pregnancy-related effects as mediator and dependent variable in the mediation model. As our main goal was to check for pregnancy-related mediators of cognitive problems in the final analysis, we used the PROCESS macro for SPSS (model 4) [50].

We asked whether we could explain the differences between pregnant and non-pregnant women in cognitive functioning by taking into account the number of awakenings. To answer this question, we performed a mediation analysis with a number of awakenings as a mediator variable and pregnancy being the main explanatory variable, and the number of remembered letters (showing the strongest effect of pregnancy) as a dependent variable.

There were no significant differences between pregnant and control women in terms of age (*p* = 0.62), education (*p* = 0.64), or verbal knowledge and concept formation skills (*p* = 0.18).

### 4.1. Comparison of Pregnant and Control Women on Cognitive Performance and Sleep Quality

In order to check for differences in cognitive functioning and sleep quality between pregnant and control women, we performed a series of independent Student’s *t*-tests for two samples. Pregnant women achieved worse results in OSPAN task (Table 1) in the overall number of remembered items (*p* = 0.12) as well as in the total OSPAN score (*p* = 0.004) than control subjects, but they did not differ in arithmetic (*p* = 0.98, Table 1).

We also found a significantly lower performance on the D2-test on the PT index, which assesses the general processing speed (*p* = 0.03, Table 1).

According to actigraphy, the sleep quality was significantly worse in pregnant women: the sleep efficiency was lower (*p* = 0.001), time of WASO was longer (*p* = 0.0001), pregnant women had a higher number of awakenings (*p* = 0.01) and longer average time of awakenings (*p* = 0.0005) than the control group. On the other hand, pregnant women spent a longer time in bed (TTB, *p* = 0.0004, Table 2). There were no significant differences in TST (*p* = 0.053, Table 2).

In the psychosocial tests performed, we found significant differences between the groups only in the score of FIRST, which was higher for pregnant women than controls (*p* = 0.02), but not for the other scales (*p* = 0.06, Table 3).

### 4.2. Quality of Sleep Explains Poor WM in Pregnant Women

In order to validate our claim that cognitive performance worsening in pregnant women might be due to lower sleep quality, we built the mediation model with pregnancy (coded as 0–1) as a main predictor (P), a cognitive functioning indexed by OSPAN task score as a dependent variable (DV) and sleep quality measured by number of awakenings as a mediator (M). Initial regressions (P→DV; P→M and M→DV) showed that all variables selected to mediation fulfilled the criteria formulated by Baron and Kenny (1986) [51]. The indirect effect of pregnancy on cognitive functioning through the average number of awakenings was statistically significant, ab = −0.28, BCa Cl [−0.41, −0.04]. The mediator accounted for roughly 40 percent of the total effect P_M_ = 0.38. The relationship between pregnancy (P) and OSPAN score (DV) was initially very strong (−0.73 **), but after introduction into the model the mediator (average number of awakenings) was significantly weaker (−0.45 **) (Figure 1).

## 5. Discussion

Our study showed that pregnant women from our research group actually had lower cognitive function scores and objectively poorer sleep quality than controls. Frequent awakening from sleep can explain poorer working memory in the pregnant women in our study.

We examined a sample of 19 pregnant women in their median 33 weeks of pregnancy and 20 non-pregnant women of similar age. Despite the fact that pregnant women in our sample obtained similar scores on verbal knowledge (WAIS) and concept formation skills to controls, they had lower scores in the overall number of remembered letters in the OSPAN test and lower scores on a general OSPAN index, which reflected a deterioration in the working memory functioning. We also found significantly lower results of the attention test D2 scores in pregnant women (*p* = 0.03). In the literature, similar results were described by Buckwalter et al. [9] and Henry and Shewrin [52].

Furthermore, the quality of sleep measured with actigraphy was significantly worse in pregnant women: they had lower sleep efficiency, longer WASO, a higher number of awakenings, and longer average time of awakenings than control non-pregnant women. On the other hand, they spent more time in bed. Objective sleep studies using polysomnography show changes in sleep quality in pregnant women, which deteriorates further with the advancement of pregnancy. In the third trimester of pregnancy a wake after sleep onset is longer and the total sleep duration is shorter [20,21,22].

Chronic insomnia is accompanied by deterioration of continuous attention, working memory, and impairment of learning [53,54]. Moreover, in insomniacs with objectively lower sleep quality as confirmed with polysomnography, working memory is more impaired than in those with objectively better sleep quality [55].

Lower sleep quality could worsen cognitive functioning during the day [56]. This is entirely explicable as sleep has beneficial effects on memory consolidation [24,57]. During post-learning, SWS memories are reactivated and stabilized, and they are reconsolidated and strengthened during REM sleep [58,59]. Working memory impairment related to sleep deprivation is significantly correlated with lower activity of the left parietal and prefrontal regions in MRI [60].

To our knowledge, the effect of sleep fragmentation on cognitive performance in pregnancy has not been studied thus far. In our study, we examined the effects of sleep worsening on working memory in pregnancy. The statistical analysis confirmed that a higher number of awakenings in the night can explain worse scores in working memory tests during the pregnancy.

Sleep hypopnea and sleep apnea could also be a reason of sleep fragmentation as well as memory decline. Up to 27% of women suffer from sleep apnea at the end of pregnancy [7]. In our study, we used the Epworth scale to assess sleepiness during the day, which is the main symptom of obstructive apnea syndrome (OSA). However, we found no significant differences between pregnant and non-pregnant women in ESS scores. In addition, we used pulse oximetry in 13 of pregnant and 11 of control women and all of them had correct parameters, similarly to the study of Sargberg et al. [44].

Despite objectively worse sleep quality, pregnant women in our sample did not complain about sleep problems and had AIS score similar to controls. They had a higher tendency to respond to stress with insomnia than controls, which has also been described in a previous study [46].

Objectively measured sleep parameters are not necessarily correlated with subjective reports. There are many publications that describe differences between the results of objective and subjective sleep measures, which show that patients with insomnia especially overestimate their problems [61,62,63]. It can therefore be assumed that pregnant women who complain about sleep problems might have greater deterioration of cognitive functioning. Our study showed that sleep fragmentation in the third trimester of pregnancy may impair working memory functioning.

Our results have several limitations. First, we did not assess sleep with polysomnography, which is the most thorough examination. We only used actigraphy and subjective measures of sleep. The results cannot be fully generalizable, because the study population was small and included only a group of pregnant women from a single clinic, located downtown in a big city; therefore, it may differ in severity of sleep disorders and psychological parameters.

Another limitation of this study is lack of ruling out drops in blood saturation in the whole group. Unfortunately, the pulse oximeter data from six pregnant and nine non-pregnant women were damaged.

The quality of sleep during pregnancy seems to be underestimated; meanwhile, it may not only have an adverse effect on mood, but also worsen cognitive functioning. Therefore, prevention and diagnosis of sleep disorders in pregnancy should be practiced. If a sleep disorder is diagnosed, cognitive behavioral therapy (CBT) for treating insomnia should be strongly recommended. The implementation of sleep hygiene rules and constant bedtime in all pregnant women, as well as those who do not report sleep disorders, can potentially improve their cognitive functioning.

It would be important to investigate the relationship between detailed sleep parameters using polysomnography and specific cognitive functions in future studies.

## 6. Conclusions

Our study showed that sleep fragmentation may be responsible for poorer cognitive functioning in late pregnancy. Therefore, improving the quality of sleep during pregnancy may not only improve the well-being of women, but it can also improve their memory.

## Figures and Tables

**Figure 1 jcm-11-05607-f001:**
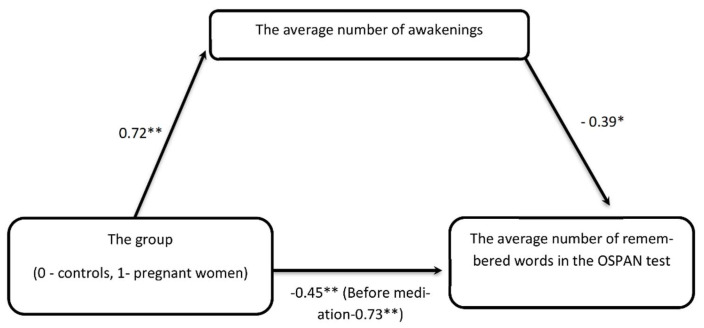
The impact of pregnancy on the OSPAN task is mediated by the average number of awakenings—the relationship between being pregnant or not, initially very strong, after introducing into the regression model of the intermediate variable in the form of the average number of awakenings is no longer statistically significant. * *p* < 0.05, ** *p* < 0.01—statistical significance differences.

**Table 1 jcm-11-05607-t001:** Cognitive performance of pregnant women and non-pregnant controls.

Group		Control	Pregnant	*t*
OSPAN task	Arithmetic	89.75 ± 5.5	89.81 ± 8.2	0.03 *
	Overall number of remembered letters	54.25 ± 13.9	42.62 ± 14.3	2.65 *
	OSPAN score (WM span index)	35.8 ± 18.8	19.62 ± 14.6	3.08 **
D2 test	PT	533.55 ± 54.1	493.52 ± 76.4	1.94 *
	Acc (%)	99.93	99.94	0.66
	CT	192.2 ± 44.6	176.67 ± 38.4	1.19
Wechsler (Vocabulary)		43.75 ± 10.15	38.62 ± 13.4	1.38

* *p* < 0.05, ** *p* < 0.01—statistical significance differences. The values shown are means, Student’s *t*-test was used.

**Table 2 jcm-11-05607-t002:** Actigraphy parameters of pregnant women and non-pregnant controls.

Group	Controls	Pregnant	*t*
Sleep Efficiency	0.92 ± 0.002	0.87 ± 0.003	4.57 ***
TTB	427.17 ± 41.6	477.11 ± 45.7	3.69 ***
TST	392.07 ± 40.3	418.71 ± 47.7	2
WASO	33.84 ± 11.3	57.33 ± 13.6	5.95 ***
Number of awakenings	12.66 ± 3.7	15.78 ± 3.7	2.6 ***
Avgerage awakenings (time, s)	2.36 ± 1.3	3.74 ± 0.9	2.52 ***

*** *p* < 0.001—statistical significance differences. The values shown are means, Student’s *t*-test was used.

**Table 3 jcm-11-05607-t003:** Psychosocial characteristics of the study population of pregnant women and non-pregnant controls.

Group	Controls	Pregnant	*t*
AIS	4.75 ± 4.75	6.58 ± 3.2	1.65
HS	35.50 ± 7.4	35.32 ± 8.0	0.81
FIRST	19.95 ± 5.6	24.11 ± 4.8	2.49 *
EPWORTH	7.75 ± 4.15	9.05 ± 4.6	0.93
BDI	5.25 ± 6.1	7.32 ± 5.15	1.14

AIS—Athens Insomnia Scale, HS—Regenstein Hyperarousal Scale, FIRST—Ford Insomnia Response to Stress, ESS—Epworth Sleepiness Scale, BDI—Beck Depression Inventory. * *p* < 0.05—statistical significance differences. The values shown are means, Student’s *t*-test was used.

## Data Availability

Data are available on request due to restrictions. The data presented in this study are available on request from the corresponding author.

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
