# Peer review of "Cognitive Function Decline in the Third Trimester of Pregnancy Is Associated with Sleep Fragmentation"

_jcm, 2022, doi:10.3390/jcm11195607_

Round 1
Reviewer 1 Report
Thank you for the opportunity to review the manuscript entitled: Cognitive functions decline in the third trimester of pregnancy is associated with sleep fragmentation. This manuscript examines whether cognitive impairment relates to sleep deterioration during the third trimester of pregnancy using a battery of cognitive assessments, psychosocial measures (i.e., depressive symptoms) and actigraphs. The authors used a series of regression models and mediation analysis to fill existing gaps in the relationship between cognitive function and sleep problems. Overall, the authors’ work contributes to better understand the role of sleep (or sleep disturbances) on cognitive functioning. There are aspects of framing, organization, methods, and results that impact the ability to properly interpret and contextualize the findings which I outline below. Main comments are presented by manuscript section with general comments at the end.
Introduction
1. If the journal allows, I suggest re-organizing the information presented in the introduction using subheadings. I would suggest organizing the introduction based on the “arms” of the mediation analysis, which seems to be the focus of the paper but is lost in the introduction. For example, starting with what is known in general regarding sleep, pregnancy and cognitive function, specific to the third trimester; then, reviewing literature on the relationship between pregnancy and cognitive function; next, reviewing literature on pregnancy and sleep; and finally, ending with a brief summary and where a mediation analysis might be necessary to triangulate and examine relationships.
2. The current study section would benefit from starting with a very brief 1-2 sentence summary of existing knowledge, then moving to discuss the aim of the current study and hypotheses.
Methods
1. Materials and Methods:
a. When comparing weight and BMI between pregnant and non-pregnant women, was pre-pregnancy weight and BMI used? Given the relationship between BMI and sleep quality (i.e., Tang, Y et al., 2022), and differences seen in the sample, was BMI adjusted for in models?
i. Tang, Y., Dai, F., Razali, N.S. et al. Sleep quality and BMI in pregnancy– a prospective cohort study. BMC Pregnancy Childbirth 22, 72 (2022). https://doi.org/10.1186/s12884-022-04414-7
2. Instruments and Procedures
a. Cognitive performance: I suggest this be streamlined and re-organized similar to the “Psychological Measurement” section.
b. A section identifying covariates (and why they were selected) is necessary to interpret findings accurately.
3. Data analysis requires more details—without this is it not possible to accurately interpret results. For example, a brief summary of exploratory, descriptive and bivariate analyses (why student’s t-test); then, details on linear regression models (multivariable or multivariate, what covariates were included if any?); and finally, details on the mediation analysis are necessary.
Results
1. Results would be better organized if the analytic plan was clearer in the methods section. Then, the results section could follow the analysis plan and provide a better flow for readers.
2. I recommend reviewing the data presented in the tables and in text to see if the authors can reduce redundancy.
3. The description of the mediation analysis should appear in the data analysis plan in the methods.
4. Recommend changing lay language (i.e., “slightly worse”) to statistical language (i.e., significantly lower) throughout the results section.
Discussion
1. Starting the discussion section with the main takeaway from the results will strengthen the implications of the findings. Then, a more detailed discussion of the results can proceed.
2. What is meant by “Our study group may differ in terms of psychopathological characteristics (i.e., more severe sleep problems, emotion)” in line 355-356?
3. There is not a clear takeaway from the results presented. Including a stronger conclusion paragraph would benefit this manuscript.
Tables/Figures
1. Please include conventional demarcation of statistical significance as indicated by * p < .05, ** p < .01, *** p < .001.
2. Consider re-organizing tables with “Control” and “Pregnant” as the column headers and the other variables as rows with variable names on the left. This will help the reader better see differences between the control and pregnant groups, as this is the focus, not the differences across tests/sleep states/psychosocial adjustment
3. Table 3. These are not psychometric characteristics of the study population. The title should be changed to reflect that these are psychosocial characteristics.
General Comments
1. Use simplified, active language throughout to help readability of manuscript

Reviewer 2 Report
The authors of this manuscript took up a very important topic in the field of mental and physical issues referred to cognitive functions decline in the third trimester of pregnancy associated with sleep fragmentation. The authors are requested to improve the manuscript and provide response to all recommendations raised by the reviewer:
-
The manuscript format is poor and does not comply to journal guidelines:
in example, no major paragraph numbering; missing “Conclusion” paragraph;
in the end of manuscript and before references section missing are:
“Author Contributions”,” Funding” information,” Institutional Review Board Statement”,
“Informed Consent Statement” , “Data Availability Statement” ,”Conflicts of Interest” Statement.
In general the manuscript appears to be formatted for a journal other than the Journal of Clinical Medicine. -
The authors do not use numerical references in the manuscript main text, but the authors' names
and probably the year of publication, i.e. line 155, line 160-161, line 176-177 . Such manner are unacceptable.
-
Line 5-7: MDs and PhDs abbreviations associated to authors' names must be removed.
-
Line 9-14: incomplete addresses of the institution. Please enter the street, Avenue or Road etc. if possible for Nowowiejska 27, Karowa 2, Pawińskiego 3, Chodakowska 13 19/31.
-
Line 35-36 (quote) “ On the other hand, in previous studies, slight objective cognitive decline in pregnant women has been found.” The quoted phrase "in previous studies" is related to earlier studies by the authors of this manuscript or others?
-
Line 113 (quote) “ Insomnia meta-analytic review (2016) reported mild deficits of working memory in insomnia subjects compared to controls [22]. “ What does "(2016)” in the quoted mean?
-
Line 131 (quote) “The study protocol was approved by the Bioethics Committee of the Medical University of Warsaw “. There is no numerical / alphabetical approval code issued by the Bioethics Committee of the Medical University of Warsaw. Therefore missing information should be updated and inserted.
-
Line 345-347 (quote) “ There are many publications that describe differences between the results of objective and subjective sleep measures, which show that especially patients with insomnia overestimate their problems [50] “. Please provide more references than one to support quoted phrase if “there are many publications”.
Round 2
Reviewer 1 Report
Thank you for taking time to incorporate feedback into this manuscript!
Reviewer 2 Report
The authors referred to the reviewer's comments and improved / revised the manuscript to a satisfactory level.